# GENERATIVE PRETRAINED EMBEDDING AND HIERARCHICAL REPRESENTATION TO UNLOCK HUMAN RHYTHM IN ACTIVITIES OF DAILY LIVING

## ABSTRACT

Within the evolving landscape of smart homes, the precise recognition of daily living activities using ambient sensor data stands paramount. This paper not only aims to bolster existing algorithms by evaluating two distinct pretrained embeddings suited for ambient sensor activations but also introduces a novel hierarchical architecture. We delve into an architecture anchored on Transformer Decoder-based pre-trained embeddings, reminiscent of the GPT design, and contrast it with the previously established state-of-the-art (SOTA) ELMo embeddings for ambient sensors. Our proposed hierarchical structure leverages the strengths of each pre-trained embedding, enabling the discernment of activity dependencies and sequence order, thereby enhancing classification precision. To further refine recognition, we incorporate into our proposed architecture an hour-of-the-day embedding. Empirical evaluations underscore the preeminence of the Transformer Decoder embedding in classification endeavors. Additionally, our innovative hierarchical design significantly bolsters the efficacy of both pre-trained embeddings, notably in capturing inter-activity nuances. The integration of temporal aspects subtly but distinctively augments classification, especially for time-sensitive activities. In conclusion, our GPT-inspired hierarchical approach, infused with temporal insights, outshines the SOTA ELMo benchmark.

## 1 INTRODUCTION

Human activity recognition (HAR) focuses on algorithms capable of finding patterns to describe human movements from sensor data. For smart homes equipped with ambient sensors (eg. motion, door open/close, temperature), taking advantage of the recent development of Internet of Things (IoT), HAR consists in recognising activites of daily living (ADL) such as cooking, sleeping, cleaning... so as to provide services for healthcare or enhance daily life.

The performance of deep learning, especially its ability to interpret raw data, has placed it at the forefront of HAR implementations in smart homes Gochoo et al. (2018); Mohmed et al. (2020); Wang et al. (2016); Singh et al. (2017). Yet, genuine challenges persist Bouchabou et al. (2021b). In particular ADL are complex actions combining several actions with multilevel dependencies in terms of time. Recognizing intricate activity sets often stumbles upon the similarities in traces and their contextual dependencies. Furthermore, the inherently privacy-respecting design of ambient sensors, which provides minimal environmental context, results in poorly informative sensory signals and thus, in time series data without the Markov property. This means that understanding one sensor activation often requires contextualization from the history and from other sensors. Activities can also be deeply interconnected, underscoring the significance of temporal context in HAR. Compounding these issues, ambient sensor data manifests as noisy, multi-variate, and irregular time series, straining traditional modeling methods.

To deal with long-term dependecy, hierarchical models have been proposed as ontologies Hong et al. (2009) or hierarchical hidden Markov models Asghari et al. (2019), all implementing the idea of multilevel dependencies in terms of time. In parallel, recent deep learning paradigms have begun to offer single-level sequence models to analyse temporal patterns Medina-Quero et al. (2018); Liciotti et al. (2019); Sedky et al. (2018). Yet, they are not yet able to cope with

irregular time series. Furthermore, handling long-range contexts remains a conundrum. Previous efforts, like leveraging Recurrent neural networks or adopting language models like ELMo Peters et al. (2018) for sensor sequences Bouchabou et al. (2021c) or GPT-2 Radford et al. (2018) for sensor sequences Takeda et al. (2023), have shown promise but hit limitations for broader contexts. Combining both streams of work in single-level time dependency and multi-level dependency, our work integrates (1) attention mechanisms for discerning importance of sensor signals across a whole sequence, (2) pre-trained generative transformer embeddings capturing sensor inter-relations, (3) a hierarchical model emphasizing activity succession for long-horizon dependency, and (4) a temporal encoding model to harness the rhythm of ADLs. In essence, we propose a multi-timescale architecture, so as to take into account a wider temporal context in a multi-timescale manner. Our code is available on `https://anonymous.4open.science/r/Generative-Pretrained-Embedding-and-Hierarchical-Representation-to-Unlock-ADL-Rhythm-in-Smart`

## 2 APPROACH

### 2.1 CAUSAL EMBEDDING

Driven by the innovative approach of pre-trained ELMo embeddings for ambient sensors in smart homes proposed by Bouchabou et al. (2021c), and the success of transformer architectures like BERT Devlin et al. (2018) and GPT Radford et al. (2018), we sought to harness the capabilities of the Transformer decoder embedding architecture to amplify the efficacy of the HAR algorithm.

Our challenge centers on the effective categorization of sensor event sequences, which are temporally contingent and intertwined, into discernible daily activities. A practical example can be visualized in the transition of a closed room from being vacant to occupied; this change naturally implies that the door had been opened prior to the occupancy. It's this cause-effect relationship that underscores the aptness of a Transformer decoder architecture, such as GPT, for our task.

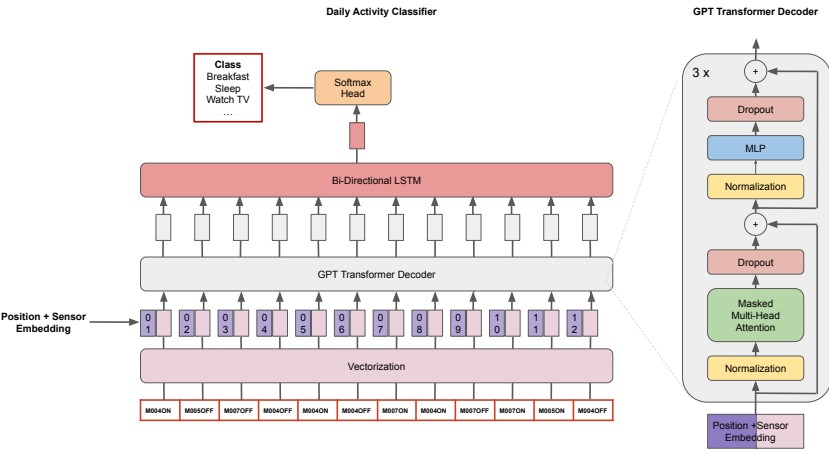

Figure 1: Model architecture of GPTAR and its GPT transformer decoder. GPTAR embeds the sensor signal with 3 layers of GPT transformer decoder embedding and a bi-LSTM. The illustration of the Transformer decoder was inspired by Vaswani et al. (2017)

In contrast, while architectures like BERT might adeptly handle intra-sequence context, their structural design constrains them from extrapolating context beyond their immediate sequence. Given the inherent causality of our challenge, the capacity of a Transformer decoder-based architecture to anticipate forthcoming events based on present sequences renders it a more fitting choice.

As illustrated in Figure 1, we propose for our single-level temporal computation integrates a pre-trained GPT Transformer decoder embedding in lieu of the earlier ELMo setup. Sensor events are encoded via this GPT embedding, which the Bi-LSTM layer then leverages to generate a cohesive vector representation. We name the classifier using this module with sofftmax layer, GPTAR.

## 2.2 MULTI-TIMESCALE ARCHITECTURE

Driven by the aspiration to discern the intricate temporal relationships both within individual activities and across successive ones, we designed a hierarchical architecture, depicted in Figure 2. Understanding that human behaviors manifest across diverse observational scales, our model is meticulously crafted to comprehend both the immediate sequences of events and the more expansive dynamics between activities.

We augmented our architecture with an additional bi-directional LSTM layer, drawing methodological inspiration from the strategies proposed in Devanne et al. (2019). This enhancement empowers our model to not only discern immediate sequences of events but also to contextualize activities within a broader framework. The bi-directionality of this LSTM layer accentuates its capability to capture and assimilate insights from both prior and subsequent events. As a result, our architecture adeptly integrates immediate event transitions with overarching behavioral patterns, furnishing a comprehensive understanding of human activity sequences.

For the input, our model processes a chronologically ordered sequence of three activities. The objective of this architectural choice is to predict the label of the current activity by leveraging the contextual representations of its two preceding activities. We opted for a sequence of three activities because, through observation, we discerned that activities often intersperse with a category termed "other", representing unlabeled sequences of sensor activations.

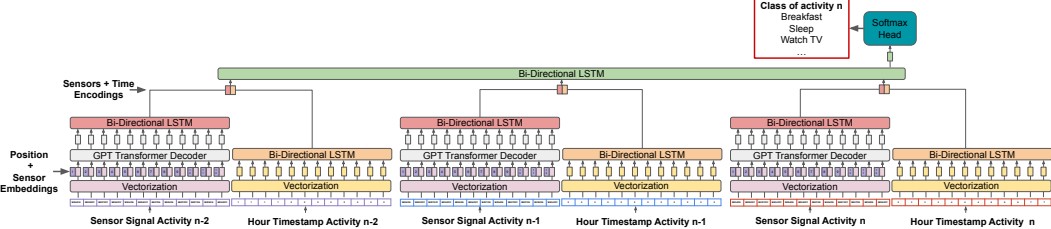

Figure 2: Complete architecture of the Generative Pre-trained Transformer for Hierarchical Activity Recognition (GPTHAR), composed of 3 low-level modules to compute 3 successive activities, and a top-level composed of a bi-LSTM and a softmax classifier. The low-level module processes in parallel the hour timestamp with a bi-LSTM and the sensor signal with a GPT transformer decoder embedding and a bi-LSTM.

## 2.3 TIME ENCODING

For smart home human activity recognition, capturing the nuances of temporal patterns is of paramount importance. Human behaviors within residential settings often exhibit rhythmic patterns, driven by ingrained habits. Therefore, understanding not just the occurrence but also the timing and sequencing of sensor-driven events can yield deeper insights. As an example, a sensor activation in the kitchen at 8 am might be indicative of breakfast preparation, while the same sensor trigger at 8 pm could be related to dinner preparations.

Building upon foundational techniques, we incorporate a specialized temporal encoding mechanism, as depicted in Figure 2. A supplementary input was introduced to our model. This input maps to the hours corresponding to each sensor activation timestamp. For every sensor activation in the main input sequence, a corresponding hour-of-the-day value is aligned in this secondary time input sequence. These hour values undergo transformation through an embedding layer, rendering them into vectorized representations. Subsequently, a bi-directional LSTM processes this sequence of vectors. The output of this LSTM is then combined with the output from the bi-directional LSTM that encodes the sensor activations. This merged output is directed to the terminal LSTM layer, designed to discern the intricate relationships and order of activities.

## 3 METHODS

### 3.1 DATASETS

To evaluate the robustness and adaptability of our model, we utilized the Aruba, Milan, and Cairo datasets from the CASAS collection Cook et al. (2012). These choices were strategic: Aruba to test single-resident scenarios, Milan to assess the impact of pets and sensor issues, and Cairo to observe performance in multi-resident settings with overlapping activities. Collected from volunteers' homes over several months, these datasets feature unbalanced classes and vary in house structures and resident numbers (details in Annex E).

### 3.2 GENERATIVE PRE-TRAINED TRANSFORMER SENSOR EMBEDDING

Following Bouchabou et al. (2021c), we use an ELMo embedding with a GPT transformer decoder Radford et al. (2018) for sensor embedding training. This approach predicts the next sensor event by analyzing the current context, similar to natural language processing techniques. Our GPT Sensor Embedding model, as shown in Figure 1, comprises token and positional embeddings, and transformer decoders with pre-normalization Xiong et al. (2020), in line with GPT-2 Radford et al. (2019). We used a context length of 1024 tokens for input contexts, matching the GPT-2 model's configuration.

### 3.3 PRE-PROCESSING, TRAINING, EVALUATION AND METRICS

Datasets are divided on a weekly basis to preserve temporal relationships. Once partitioned, the weeks are shuffled and divided into training (70%) and testing (30%) subsets. The training subset is multifunctional, facilitating the training of pre-trained embeddings, hyperparameter optimization via cross-validation, and the training of the classification model.

In the embedding pre-training step, 80% of the training subset is designated for model training, with the remaining 20% allocated for validation. We employ early stopping, based on validation perplexity, to counteract overfitting.

For hyperparameter tuning, a 3-fold cross-validation approach is adopted. From the first two folds, 20% is reserved for validation and early stopping. The third fold serves as the test set for cross-validation. In the final classification step, 20% of the whole training subset is earmarked for validation and early stopping. Subsequently, the algorithm undergoes testing on the test set defined initially.

Given the significant imbalance in our activity classes, we report the F1-score metric. For statistical significance, we report the test results averaged over 10 repetitions. For experiment reproducibility, we use a fixed value for all random seeds.

### 3.4 VECTORIZATION OF SENSOR ACTIVATIONS

In our approach, sensor activations are turned into categorical symbols, helping the model identify patterns and relationships. This forms a vocabulary that encapsulates the nuances of sensors' activations. Our datasets include motion (M), door (D), and temperature (T) sensors. Every event is logged with details including a unique sensor ID, its corresponding value, and a timestamp.

We convert each event, defined by sensor ID ($s_i$), value ($v_i$), and timestamp ($t_i$), into a unique token by merging $s_i$ and $v_i$, excluding $t_i$. For example, motion sensor M001 turning ON becomes 'M001ON', and temperature sensor T004 reading $24.5°C$ becomes 'T00424.5'.

For processing, these tokens are indexed similarly to methods used in natural language processing. This means that index assignment is based on frequency, starting from 1, with 0 reserved for padding. Consequently, a sequence such as [M005OFF M007OFF M004OFF M004ON] is transformed into an indexed sequence like [1 4 8 2], in accordance with the frequency of each token.

## 4 RESULTS

In this section we provide an empirical study of the following research questions: (RQ1) Can GPT-based models capture better long-term dependencies and lead to better activity recognition ? (RQ2) Is a single-level long-term dependency enough to model activities of daily living and what does a hierarchical model add ? (RQ3) Is time a relevant information for HAR, especially in the case of this irregularly sampled time-series from event-triggered sensors ?

To this end, we consider ablation studies with the following algorithms : (i) GPTHAR, our proposed method, using GPT transformer decoder with time-encoding and a hierarchical architecture as pictured in Fig. 2. (ii) GPTAR-note (for *no temporal encoding*) : uses GPT transoformer decoder and a hierarchical architecture, but without timestamp information . (iii) GPTAR, pictured in Figure 1, using GPT transformer decoder for embedding in a single-level architecture, without timestamp information. (iv) ELMoHAR : using ELMo with time-encoding and a hierarchical architecture. (v) ELMoAHR-note (for no temporal encoding) : uses ELMo and a hierarchical architecture, but without timestamp information . (vi) ELMoAR : using ELMo for embedding in a single-level architecture, without timestamp information.

### 4.1 COMPARATIVE STUDY AND HYPERPARAMETERS SEARCH

To respond to RQ1, we compare two distinct embeddings: the GPT decoder Transformer-based model (GPTAR) and the ELMo-based model (ELMoAR) as described in Bouchabou et al. (2021c). These models were implemented according to the architecture delineated in Fig. 1, where the embedding component is implemented with ELMo embedding for ELMoAR and GPT decoder Transformer embedding for GPTAR. Both embeddings underwent pre-training on the training set : GPT decoder on unsegmented data, and EMLo on pre-segmented data.

A hyperparameter tuning was conducted for each embedding technique, encompassing variations in window size for ELMoAR and the number of layers and attention heads for GPTAR. The outcomes of a 3-fold cross-validation on the three datasets are reported in Table 4 in Annex A for the F1 score and in Annex B for more metrics. For ELMoAR, a 60-token context window offers the best results across all datasets. For GPTAR, a configuration with 8 attention heads and 3 decoder layers yielded better F1 scores averaged across the three datasets, in particular the noisy Milan and Cairo datasets, and exhibiting less variance then ELMoAR. The augmentation of attention heads or decoder layers in the GPTAR model did not linearly correlate with improved recognition performance. GPTAR shows better robustness in noisy datasets and an improved average F1 score during cross-validation.

However, an in-depth analysis of individual activity performance in Annex C and confusion matrix in Annex D revealed specific challenges. Activities such as 'Wash Dishes' in Aruba, 'Eve Meds' in Milan, and 'Breakfast' in Cairo, presented suboptimal recognition. These activities, often exhibiting similar sensor patterns, however occur after different activities, in the perspective of a sequence of activities.

The next sections use on these two hyperparameters for low-level modules of ELMoAR, ELMo-HAR-note, ELMoHAR, GPTAR, GPTHAR-note and GPTHAR: the 60-token window for ELMo and the 8 attention heads and 3 decoder layers for GPT embedding.

### 4.2 HIERARCHICAL ACTIVITY RECOGNITION

ADLs often manifest interconnected patterns, driven by established daily routines or a structured sequence. While deciphering the relationships among individual sensor activations, comprehending the overarching dynamics between the activities, in particular the multi-timescale dependencies, is ncessary. To respond to RQ2, we examine how a single-level model, even understanding a long time-range dependencies, compares to a hierarchical model.

We delve into the hierarchical architectures, Generative Pre-Trained Hierarchical Activity Recognition - no time encoding (GPTHAR-note) and ELMo Hierarchical Activity Recognition - no time encoding (ELMoHAR-note), built upon GPTAR and EMLoAR to encapsulate these inter-activity relationships. They differ from Fig. 2 in the absence of the timestamp input. We compare the hierarchical and non-hierarchical architectures in section 4.2.1. To test the hypothesis that the hierarchical

architectures benefit from data from longer time ranges and from activity segmentation information, we compared with the non-hierarchal algorithms with longer input context and with a segmentation sign (section 4.2.2).

### 4.2.1 COMPARISON OF HIERARCHICAL ARCHITECTURES

In our experiments with GPTHAR and ELMoHAR, we employed the previously selected ELMo-based model with a 60-token context window and the GPT transformer decoder embedding model configured with 8 attention heads and 3 decoder layers. Consistent with earlier experiments, the pre-trained embeddings remained fixed, allowing only the dual bi-directional LSTM layers and the softmax layer to undergo training for classification. Comparative results from these experiments are presented in Table 1. We also compared to other families of methods: FCN Bouchabou et al. (2021a) and a SOTA LSTM-based method Liciotti et al. (2019). For details of Liciotti et al. (2019) replication, please refer to Annex F.

Table 1: Hierarchical model : Test F1 score of FCN, Liciotti et al, and the hierarchical and non-hierarchical architectures using ELMo and GPT pre-trained embeddings.

|  | Aruba | | Milan | | Cairo | |
|---|---|---|---|---|---|---|
|  | Macro F1 Score | std | Macro F1 Score | std | Macro F1 Score | std |
| FCN Bouchabou et al. (2021a) | 33.10% | 2.23 | 15.10 | 1.52 | 7.60% | 2.46 |
| Liciotti et al. Liciotti et al. (2019) | 32.00% | 1.56 | 17.40% | 2.07 | 26.60% | 3.24 |
| ELMoAR (Window 60) | 84.80% | 1.99 | 70.80% | **0.92** | 70.50% | **1.43** |
| GPTAR (8 Heads 3 Layers) | 86.10% | **1.20** | 70.80% | 1.40 | 73.20% | 1.99 |
| ELMoHAR-note | **88.10%** | **1.20** | 77.40 | 1.65 | 75.90% | 2.88 |
| GPTHAR-note | 87.30% | 2.98 | **79.90%** | 1.52 | **84.80%** | 1.81 |

The results indicate that GPTHAR-note outperforms all other architectures across the three datasets. It can be observed that the hierarchical structure allows both types of pre-trained embeddings to enhance their performance. This underscores the presence of relationships among activity sequences, which significantly bolsters the model's classification performance. However, it's worth noting that for the Aruba and Milan datasets, employing a hierarchical structure increases the standard deviation value. In these datasets, the model displays reduced stability in terms of consistent performance. We hypothesize that introducing a regularization layer after the embedding output might help stabilize the architecture's performance. Upon a detailed examination of classification performances for individual activities across the three datasets (see Annex C and D), integrating relational context among activities improves recognition rates. For instance, activities that depend on preceding ones, such as "Wash dishes" – which is directly associated with activities like "Dining Room Activity" or "Meal Preparation" in the Aruba dataset – benefit from this approach. Similarly, activities like "Leave home" and "Enter Home", which often get misclassified, are better distinguished using the hierarchical architecture.

### 4.2.2 INPUT CONTEXT EXTENDED

In assessing the hierarchical structure's efficacy, we contrasted hierarchical and non-hierarchical algorithm versions using an augmented input context. Specifically, base models ELMoAR and GPTAR inputs were expanded by: 1) appending two preceding activities to the current one, retaining sensor event sequence, and 2) appending two preceding activities with a separation token between distinct sensor sequences. This token demarcates activity boundaries, inherently discerned by GPTHAR and ELMoHAR. Results are presented in Table 2.

The results demonstrate that the hierarchical structures maintain superior performance, compared to the versions of the models with a extended input context. We hypothesized that since the hierarchical structure inherently discerns the boundaries between activities through its design, the addition of an explicit token to mark this boundary might suffice to achieve equivalent performance. However, we observe this is not the case. Despite marking the boundaries between activities, hierarchical structures achieve the best performance across the three datasets. It is worth noting that the versions with separators outperform their counterparts without separators. During our experiments, we observed that by increasing the number of nodes into the last bi-directional LSTM of the baseline models, both the extended context and extended context with separator approaches enhanced their performance. Yet, even by increasing the number of parameters, we could not match the performance of

Table 2: Long-term dependency : Test F1 Score and its standard deviation of the classification when using either (1) a hierarchical architecture, or (2) the baseline models using extended input context, or (3) extended input context incorporating the "$< sep >$" token.

| | Aruba | | Milan | | Cairo | |
|---|---|---|---|---|---|---|
| | Macro F1 Score | std | Macro F1 Score | std | Macro F1 Score | std |
| ELMoHAR-note | **88.10%** | **1.20** | 77.40% | 1.65 | 75.90% | 2.88 |
| GPTHAR-note | 87.30% | 2.98 | **79.90%** | 1.52 | **84.80%** | **1.81** |
| ELMoAR (Window 60) context entended | 76.90% | 1.2 | 60.05% | 3.5 | 59.10% | 3.84 |
| ELMoAR (Window 60) context extended with $< sep >$ | 77.90% | 3.63 | 59.50% | 1.27 | 56.80% | 5.45 |
| GPTAR (8 Heads 3 Layers) context extended | 81.00% | 3.53 | 61.80% | 1.4 | 58.80% | 2.7 |
| GPTAR (8 Heads 3 Layers) context extended with $< sep >$ | 81.60% | 3.81 | 61.40% | **1.26** | 60.20% | 2.94 |

the hierarchical structures. This experiment clearly indicates a need to observe sensor activations at different scales, and the proposed hierarchical structure facilitates these multi-scale observations.

### 4.3 THE IMPACT OF TIME ENCODING

Human activity can also be linked to specific times of the day. Certain activities occur at distinct moments, such as having breakfast, lunch, dinner, or sleeping. Thus, we added a temporal encoding to our activity sequences, as shown in Figure 2. The comparative results of the models, both with (ELMoHAR and GPTHAR) and without (ELMoHAR-note and GPTHAR-note) this time encoding, are presented in Table 3 (see Annex B for more metrics).

Table 3: Time encoding : Test F1 Score and its standard deviation comparing when the embedding is either (1) ELMoHAR, (2) GPTHAR with and without Time Encoding.

| | Aruba | | Milan | | Cairo | |
|---|---|---|---|---|---|---|
| | Macro F1 Score | std | Macro F1 Score | std | Macro F1 Score | std |
| ELMoHAR-note | 88.10% | **1.20** | 77.40% | 1.65 | 75.90% | 2.88 |
| GPTHAR-note | 87.30% | 2.98 | 79.90% | 1.52 | 84.80% | 1.81 |
| ELMoHAR | **90.70%** | 1.25 | 80.00 | 1.33 | 83.30% | 2.95 |
| GPTHAR | 89.70% | 3.06 | **81.90%** | **1.1** | **87.20** | **0.92** |

The results demonstrate that the integration of the temporal component augments the classification efficacy of both models. Furthermore, the addition of this temporal dimension tends to diminish the standard deviation values. This progression is particularly evident in GPTHAR-note. We note that GPTHAR-note outstrips the performance of ELMoHAR-note and ELMoAR on the Cairo dataset. On the Milan dataset, GPTHAR also considerably surpasses ELMoHAR and ELMoAR. Examining the confusion matrices in Annex D, in the Aruba dataset, activities such as 'Washing Dishes', 'Meal Preparation', 'Enter Home', and 'Leave Home' show improved classification accuracy. In the Milan dataset, 'Eve Med' and 'Morning Meds' demonstrate a notable reduction in misclassifications. Similarly, in the Cairo dataset, meal-related activities like 'Breakfast', 'Lunch', and 'Dinner' are identified with higher accuracy.

The Cairo dataset, given its complexity stemming from activities of multiple residents, shows noteworthy enhancement compared to the second noisy dataset Milan. This performance variance can be rationalized by the Milan dataset's inherent challenges. Despite seeming less complex, the Milan dataset is fraught with sensor anomalies, particularly at its onset, leading to elevated data noise.

In summary, our results that GPT-based transformers provide a richer embedding for HAR, that time encoding based on the hour of the day time can alleviate confusion between some activites (ex confusion between dinner,lunch and breakfast, see Annex C and D) for the details), and that a hierarchical model is essential despite long-term dependency embedding and segmentation information.

## 5 RELATED WORKS

### 5.1 PRE-TRAINED EMBEDDINGS

Deep learning, particularly in applications like computer vision and natural language processing, has significantly progressed by abstracting complex data (Pouyanfar et al., 2018; Ordóñez & Roggen,

2016). In HAR, sensor-based deep learning techniques have been thoroughly investigated Wang et al. (2019), with methods categorized into Convolutional Neural Networks (Singh et al., 2017; Mohmed et al., 2020), autoencoders (Wang et al., 2016), semantics-based approaches Yamada et al. (2007), and sequence models (Ghods & Cook, 2019), according to Bouchabou et al. (2021b). Yet, these often struggle with temporal aspects and long-term dependencies crucial for ADL.

Recent self-supervised learning and sequence modeling advancements have impacted HAR in smart homes, seen in the use of bi-LSTM (Liciotti et al., 2019), ELMo (Bouchabou et al., 2021c), and GPT-2 (Takeda et al., 2023) embeddings. Bouchabou et al. (2021c) enhanced bi-LSTM models with a pre-trained, frozen sensor embedding, differing from Liciotti et al. (2019)'s in-training embedding learning. Takeda et al. (2023) used a GPT embedding for sensor event prediction. However, their prediction needs the outputs from a distinct classification process and from a district segmentation module, for which they applied feature engineering, varying the methods for different datasets, instead of an end-to-end modelling.

Our work aligns with Takeda et al. (2023) in using the GPT-2 transformer's decoder block but differs by training the embedding from actual continuous sensor readings, without separate segmentation and classification modules. Unlike Takeda et al. (2023), we integrate the GPT-based embedding into a sequence representation and feature extraction architecture as in Bouchabou et al. (2021c).

## 5.2 HIERARCHICAL MODEL OF ACTIONS

Our results show that long-term dependency is not enough, but a hierarchical description is essential for modeling human actions. This article proposes yet another hierarchical model to describe complex actions. Indeed, for human activity recognition, hierarchical models have been proposed as ontologies of context-aware activities for recognition of activities of daily living in smart homes Hong et al. (2009), hierarchical hidden Markov models Asghari et al. (2019) for recognition of activities of daily living in smart homes, or hierarchical LSTM with two hidden layers for activity recognition from wearable sensors Wang & Liu (2020) or a hierarchical LSTM for activity recognition for simple action recognition from RGB-D videos Devanne et al. (2019). While Wang & Liu (2020); Devanne et al. (2019) is applied on other types of input data and other categories of activities, Hong et al. (2009); Asghari et al. (2019) focus on the same application case. While Hong et al. (2009) study user-designed models of ontology and Asghari et al. (2019) uses a HMM which does have long-term memory, our proposition uses self-attention models which are the state of the art for recognising long-term dependencies.

Besides, contrarily to the cited works, our study includes a systematic comparison of hierarchical and non-hierarchical architectures between several encodings : ELMo and GPT. From the previously cited works, only Devanne et al. (2019) compared their hierarchical structure to the non-hierarchical structure, to draw the same conclusion. Our study also reject the hypothesis that the hierarchical algorithms simply receive longer the gain only comes from processing more information from inputs from longer time-windows or from segmentation of activities.

These results only highlight the main challenge of recognition of activities of daily living : multilevel time dependencies. Indeed, an activity of daily living such as cooking and cleaning can vary greatly from one day to the other depending on the context and goal of the inhabitant. Each activity can be viewed as a combination of unit actions (such as actions recognized in Devanne et al. (2019); Wang & Liu (2020) that are selected and organised for the completion of a temporally distant goal.

While the hypothesis that complex actions need to be represented by hierarchical models is valid for classification, the literature consolidates these results for generative models and especially reinforcement learning : to solve complex tasks, hierarchical reinforcement learning (Barto & Mahadevan, 2003; Barto et al., 2013) enabled tackling complex tasks by decomposing into subtasks. These machine learning models confirm the neuroscience description that our nervous system selects and organizes motor elements in a hierarchical model Grafton & de C. Hamilton (2007), as well as the behavioural psychology studies such as Eckstein & Collins (2021) showing that humans use hierarchical representations of action sequences of efficient planning and flexibility. Our study provides yet another computation model supporting this hypothesis of hierarchy, but has the particularity of analysing activities of daily living which are very complex and variable tasks.

# 6 DISCUSSION

## 6.1 SUMMARY

In our study on recognizing daily living activities through ambient sensor classifications within smart homes, we embarked on a comparison of two distinct pretrained embeddings applied to ambient sensor activations. Notably, the Transformer Decoder-based embedding (akin to the GPT design) was shown superior in classification tasks when compared to the ELMo pretrained embeddings. By introducing a hierarchical structure, we aimed to exploit the inherent relationships among activity sequences, thereby refining the classification outcomes. The effectiveness of this approach was evident across all three datasets, with the GPTHAR version standing out especially. Furthermore, the inclusion of an hour-of-the-day embedding subtly yet significantly enhanced classification precision, particularly for activities with time-sensitive natures.

## 6.2 LIMITATIONS

Our study provides valuable insights into the recognition of daily living activities within smart homes. However,utilizing pre-segmented data may not effectively capture the natural flow of sensor activations, thereby possibly not mirroring real-world situations accurately. Such segmentation restricts the potential application of these algorithms in real-time services. Furthermore, managing large datasets introduces challenges related to training durations. Throughout our experiments, we observed that the transformer architecture resulted in longer training times compared to the LSTM embedding-based architecture. This necessitates further optimization efforts to prevent excessive training durations. Additionally, although we selected datasets that encompass various lifestyle configurations, our evaluation was limited to just three datasets. This constraint raises potential concerns regarding the wider applicability and generalizability of our conclusions.

## 6.3 FUTURE WORKS

In light of the results and observed limitations, several avenues present themselves for future exploration. Enhancing model stability stands as a priority, and we believe the introduction of normalization layers could potentially address this. Our findings from the increased standard deviation values in datasets, like Aruba and Milan, upon employing a hierarchical structure, underscore the necessity of such improvements. An exhaustive hyperparameter search is also on the horizon, aiming not only to bolster results but also to refine and make the model more efficient. Such optimization would be instrumental in achieving better performance while ensuring computational efficiency. Broadening the scope of our evaluations to encompass a more diverse range of datasets will be pivotal. This would not only test the model's robustness but also enhance its generalizability across various scenarios and environments. Lastly, we see a promising avenue in automated segmentation learning. Moving away from pre-segmented data, investigating methods that allow for more natural and continuous activity recognition could pave the way for more realistic and adaptive models. This would potentially overcome the constraints posed by our current segmentation approach, leading to more organic and real-world applicable results.

## 6.4 CONCLUSION

In the context of smart homes, our study in recognition of daily living activities underscores the significant advantages of utilizing pre-trained embeddings, with a particular emphasis on the Transformer Decoder-based approach, for this task. We introduced a robust framework dedicated to enhancing the recognition of ADLs. Our results consistently support the hierarchical methodology, highlighting its proficiency in discerning inter-activity relations. With the integration of temporal data, the framework's performance was notably augmented, especially in datasets characterized by sensor anomalies and noise, demonstrating its potential in distinguishing nuanced activity patterns.

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

# A    HYPER PARAMETER SEARCH

In this annex, we present F1 scores during cross-validation of ELMoAR and GPTAR with, respectively, different window sizes and different head and layer numbers. The best hyperparameters are selected for the ELMoHAR-note, ELMoHAR and GPTHAR-note and GPTHAR algorithms.

For ELMoAR, the best hyperparameters are: a window size of 60. For GPTAR, the best hyperparameters are : 8 attention heads and 3 layers of decoder.

Table 4: Cross-validation F1-Scores with their standard deviation for GPTAR and ELMoAR models with various hyperparameters across the three datasets, to assess the Impact of context window size in ELMoAR and the number of layers and attention heads in GPTAR

|  | Aruba | | Milan | | Cairo | | |
|---|---|---|---|---|---|---|---|
|  | Macro F1 Score | std | Macro F1 Score | std | Macro F1 Score | std | Average |
| ELMoAR (Window 20) | 83.47 | 1.83 | 71.10 | 2.25 | 66.37 | 3.24 | 73.65 |
| ELMoAR (Window 40) | 82.93 | 2.12 | 70.73 | 2.43 | 66.70 | 4.07 | 73.45 |
| ELMoAR (Window 60) | 83.47 | 2.61 | *72.40* | *2.49* | *67.23* | *4.70* | *74.37* |
| GPTAR (8 Heads 3 Layers) | 83.20 | 1.45 | **73.77** | **2.19** | **70.90** | **3.48** | **75.96** |
| GPTAR (8 Heads 4 Layers) | 83.53 | 1.48 | 72.30 | 1.62 | 69.03 | 4.42 | 74.95 |
| GPTAR (12 Heads 6 Layers) | **83.57** | **1.57** | 73.07 | 2.13 | 68.27 | 4.08 | 74.97 |

# B  DETAILED ALGORITHMS METRICS

In this annex, we present a comprehensive breakdown of the performance metrics for the algorithms used in our study. The table encompasses test results from three distinct datasets: Aruba (Table 5), Milan (Table 6), and Cairo (Table 7). These metrics show that for the most simple dataset, Aruba, ELMoHAR and GPTHAR perform closely, ranking first or second depending on the chosen metric. For the more complex datasets, Milan and Cairo, GPTHAR outperforms all the other algorithms, regardless of the choice of metric.

Table 5: Detailed Algorithms Scores over the datasets Aruba

|  | ELMoAR | GPTAR | ELMoHAR-note | GPTHAR-note | ELMoHAR | GPTHAR |
|---|---|---|---|---|---|---|
| Accuracy | 97.00% | 97.10% | 98.40% | 98.20% | 98.50% | **98.52%** |
| Precision | 86.00% | 90.30% | 89.90% | 87.40% | **94.30%** | 91.20% |
| Recall | 84.70% | 85.10% | 88.10% | 88.60% | 89.70% | **89.80%** |
| F1 Score | 84.80% | 86.10% | 88.10% | 87.30% | **90.70%** | 89.70% |
| Balanced Accuracy | 84.76% | 85.18% | 88.22% | 88.55 | 89.71% | **90.10%** |
| Weighted Precision | 96.70% | 97.00% | 98.10% | 98.00% | **98.50%** | 98.00% |
| Weighted Recall | 97.00% | 97.00% | 98.40% | 92.20% | **98.50%** | 98.30% |
| Weighted F1 Score | 96.9% | 97.00% | 98.00% | 98.00% | **98.30%** | 98.20% |

Table 6: Detailed Algorithms Scores over the dataset Milan

|  | Milan | | | | | |
|---|---|---|---|---|---|---|
|  | ELMoAR | GPTAR | ELMoHAR-note | GPTHAR-note | ELMoHAR | GPTHAR |
| Accuracy | 87.50% | 88.20% | 90.00% | 91.30% | 90.60% | **91.9%** |
| Precision | 75.90% | 80.20% | 85.60% | 87.60% | 88.90% | **90.00%** |
| Recall | 68.40 | 68.50% | 73.90 | 76.90% | 75.60% | **79.20%** |
| F1 Score | 70.80% | 70.80% | 77.40% | 79.90% | 80.00% | **81.90%** |
| Balanced Accuracy | 68.51% | 68.55% | 73.91% | 76.87% | 77.84% | **79.22%** |
| Weighted Precision | 86.80% | 88.20% | 89.60% | 91.20% | 90.70% | **91.90%** |
| Weighted Recall | 87.50% | 88.20% | 90.00% | 91.30% | 90.60% | **91.90%** |
| Weighted F1 Score | 86.70% | 87.60% | 89.20% | 90.70% | 90.00% | **91.70%** |

Table 7: Detailed Algorithms Scores over the dataset Cairo

|  | Cairo | | | | | |
|---|---|---|---|---|---|---|
|  | ELMoAR | GPTAR | ELMoHAR-note | GPTHAR-note | ELMoHAR | GPTHAR |
| Accuracy | 81.1% | 83.40% | 87.30% | 91.00% | 90.80% | **93.20** |
| Precision | 72.70% | 76.60% | 79.70% | 87.40% | 87.30% | **89.80%** |
| Recall | 69.20% | 71.40% | 74.70% | 83.40% | 82.00% | **86.60%** |
| F1 Score | 70.50% | 73.20% | 75.90% | 84.80% | 83.30% | **87.20%** |
| Balanced Accuracy | 69.12% | 71.33% | 74.75% | 83.58% | 81.87% | **86.74%** |
| Weighted Precision | 81.10% | 82.40% | 86.90% | 91.10% | 90.60% | **93.20** |
| Weighted Recall | 81.10% | 83.40% | 87.30% | 91.00% | 90.80% | **93.20** |
| Weighted F1 Score | 80.90% | 82.30% | 86.80% | 90.50% | 90.30% | **92.70%** |

## C    F1 Score By Activities for each algorithm

We report here the F1-scores as recognition performance for the 3 datasets per algorithm and per activity label, as labelled in each dataset. This annex gives an insight which activities could account for the difference of performance.

Table 8: F1-Score by activity for each algorithm on the Aruba dataset

|  | ELMoAR | GPTAR | ELMoHAR-note | GPTHAR-note | ELMoHAR | GPTHAR |
|---|---|---|---|---|---|---|
| **Bed_to_Toilet** | 0,987 | 0,996 | 0,991 | 0,992 | 0,992 | **0,996** |
| **Eating** | 0,925 | 0,935 | 0,937 | **0,948** | 0,932 | 0,936 |
| **Enter_Home** | 0,8 | 0,797 | 0,992 | **0,994** | 0,99 | 0,992 |
| **Housekeeping** | 0,83 | 0,84 | 0,829 | 0,907 | 0,852 | **0,918** |
| **Leave_Home** | 0,827 | 0,811 | **0,992** | **0,992** | 0,99 | 0,99 |
| **Meal_Preparation** | **0,974** | 0,971 | 0,972 | 0,968 | 0,971 | 0,964 |
| **Other** | 0,988 | 0,99 | 0,99 | 0,99 | 0,99 | 0,99 |
| **Relax** | 0,99 | 0,99 | 0,991 | 0,993 | **0,994** | 0,992 |
| **Respirate** | 0,867 | **0,967** | 0,651 | 0,502 | **0,967** | 0,548 |
| **Sleeping** | 0,988 | 0,99 | 0,982 | 0,984 | **0,991** | 0,981 |
| **Wash_Dishes** | 0,008 | 0,047 | **0,266** | 0,243 | 0,225 | 0,199 |
| **Work** | 0,984 | **0,993** | 0,979 | 0,987 | 0,985 | 0,975 |

Table 9: F1-Score by activity for each algorithm on Milan dataset

|  | ELMoAR | GPTAR | ELMoHAR-note | GPTHAR-note | ELMoHAR | GPTHAR |
|---|---|---|---|---|---|---|
| **Bed_to_Toilet** | 0,551 | 0,532 | 0,79 | 0,749 | 0,845 | **0,902** |
| **Chores** | 0 | 0,011 | 0,088 | 0,122 | 0,15 | **0,161** |
| **Desk_Activity** | 0,976 | **0,996** | 0,952 | 0,982 | 0,958 | 0,976 |
| **Dining_Rm_Activity** | 0,416 | 0,252 | 0,516 | 0,481 | 0,491 | **0,522** |
| **Eve_Meds** | 0 | 0,095 | 0,421 | **0,587** | 0,566 | 0,545 |
| **Guest_Bathroom** | 0,978 | 0,981 | 0,979 | 0,98 | 0,98 | **0,984** |
| **Kitchen_Activity** | 0,91 | 0,919 | 0,93 | **0,936** | 0,927 | 0,931 |
| **Leave_Home** | 0,901 | 0,911 | 0,934 | 0,952 | 0,941 | **0,957** |
| **Master_Bathroom** | 0,884 | 0,858 | 0,942 | 0,943 | 0,968 | **0,977** |
| **Master_Bedroom_Activity** | 0,788 | 0,792 | 0,799 | 0,836 | 0,836 | **0,872** |
| **Meditate** | 0,86 | 0,874 | 0,863 | 0,933 | 0,866 | **0,972** |
| **Morning_Meds** | 0,58 | 0,546 | 0,642 | 0,669 | 0,698 | **0,712** |
| **Other** | 0,888 | 0,9 | 0,906 | 0,92 | 0,909 | **0,924** |
| **Read** | 0,92 | 0,946 | 0,917 | 0,947 | 0,914 | **0,953** |
| **Sleep** | 0,897 | 0,921 | 0,927 | 0,925 | **0,943** | 0,932 |
| **Watch_TV** | 0,767 | 0,778 | 0,782 | **0,807** | 0,789 | 0,806 |

Table 10: F1-Score by activity for each algorithm on Cairo dataset

|  | ELMoAR | GPTAR | ELMoHAR-note | GPTHAR-note | ELMoHAR | GPTHAR |
|---|---|---|---|---|---|---|
| **Bed_to_toilet** | 0,382 | 0,359 | 0,374 | 0,483 | **0,545** | 0,311 |
| **Breakfast** | 0,533 | 0,61 | 0,822 | 0,89 | 0,886 | **0,94** |
| **Dinner** | 0,415 | 0,412 | 0,711 | 0,713 | 0,961 | **0,991** |
| **Laundry** | 0,911 | 1 | 0,309 | **0,951** | 0,358 | 0,794 |
| **Leave_home** | 0,938 | 0,958 | 0,891 | 0,97 | 0,895 | **0,973** |
| **Lunch** | 0,337 | 0,331 | 0,609 | 0,67 | 0,912 | **0,929** |
| **Night_wandering** | 0,759 | 0,795 | 0,805 | 0,829 | 0,823 | **0,837** |
| **Other** | 0,915 | 0,923 | 0,948 | 0,968 | 0,955 | **0,977** |
| **R1_sleep** | 0,702 | 0,629 | 0,852 | 0,868 | **0,897** | 0,877 |
| **R1_wake** | 0,871 | 0,899 | 0,903 | 0,906 | 0,895 | **0,909** |
| **R1_work_in_office** | 0,853 | 0,939 | 0,918 | 0,985 | 0,92 | **0,991** |
| **R2_sleep** | 0,706 | 0,715 | 0,824 | 0,857 | 0,856 | **0,863** |
| **R2_take_medicine** | 0,781 | 0,913 | 0,859 | 0,94 | 0,87 | **0,931** |
| **R2_wake** | 0,745 | 0,746 | 0,799 | 0,821 | 0,852 | **0,87** |

# D CONFUSION MATRIX

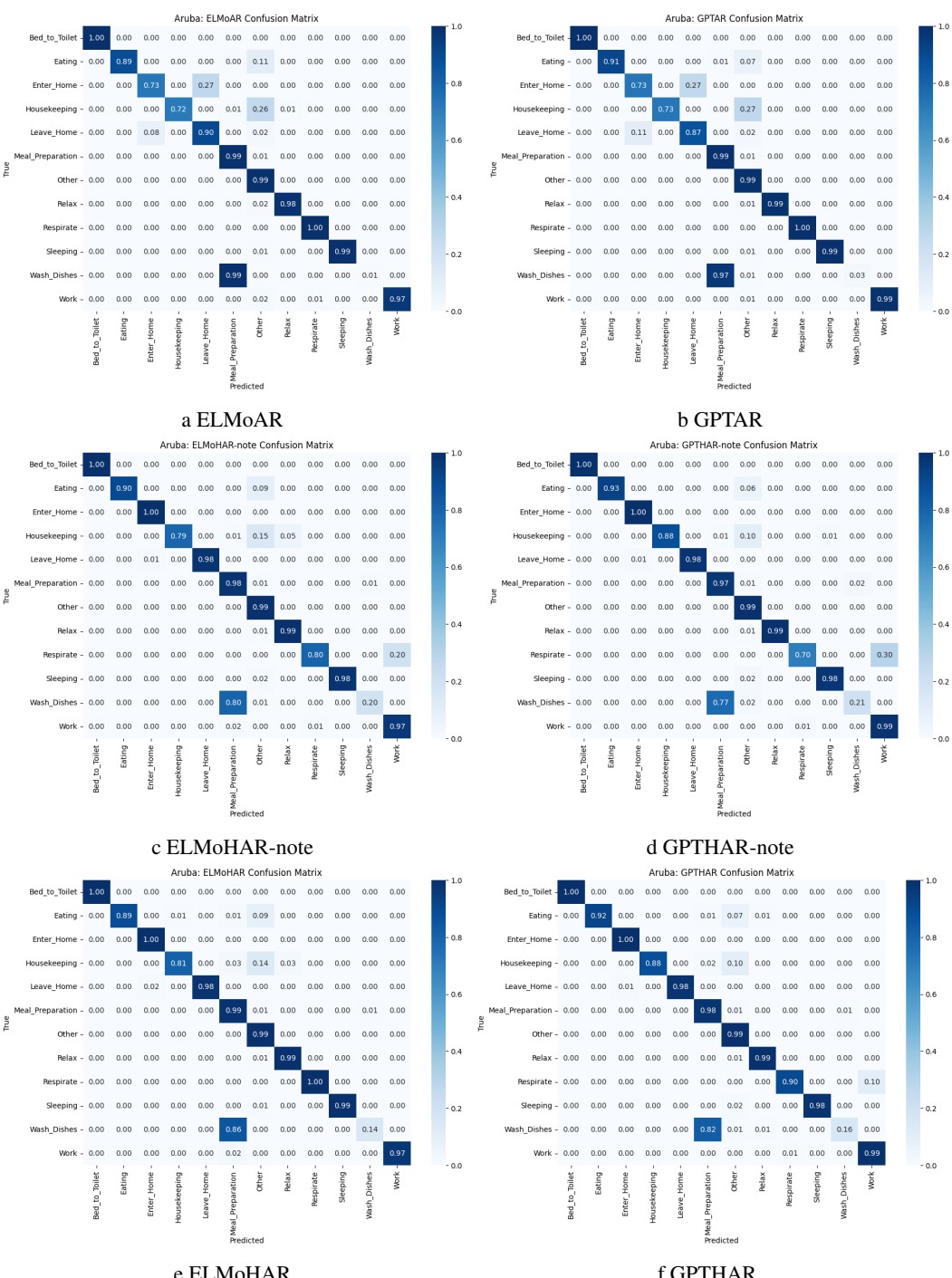

Figure 3: Confusion matrices per algorithm on the Aruba dataset.

In this section, we report on the confusion matrices for various algorithms across three datasets. A notable observation is that the more complex architectures, which include time encodings (namely ELMoHAR and GPTHAR), exhibit fewer misclassifications compared to simpler models.

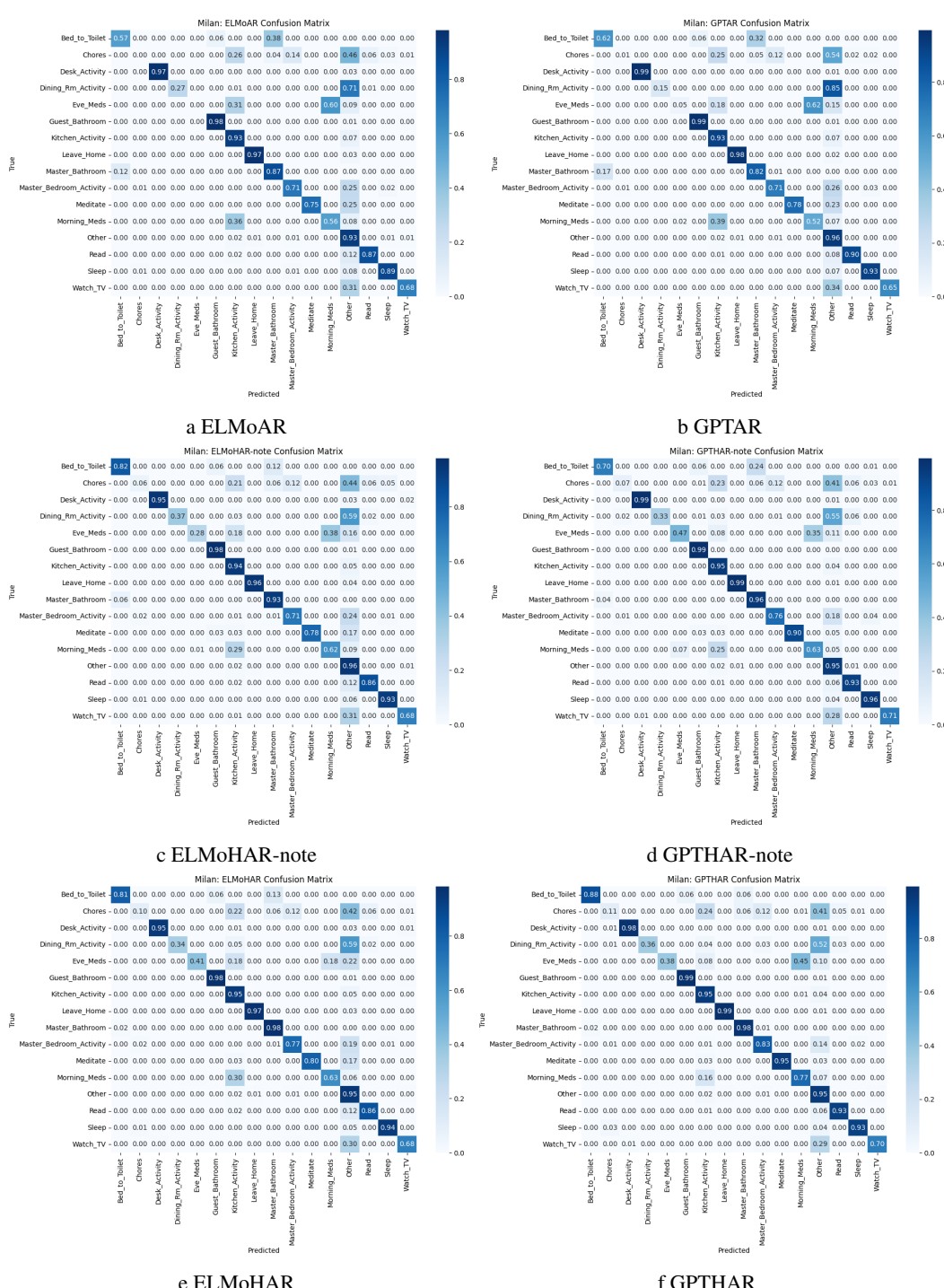

Figure 4: Confusion matrices per algorithm on the Milan dataset.

This improvement is particularly evident in specific activities. For instance, in the Aruba dataset, activities like 'Washing Dishes', 'Meal Preparation', 'Enter Home', and 'Leave Home' are classified more accurately. Similarly, in the Milan dataset, activities such as 'Eve Med' and 'Morning Meds' show a marked decrease in misclassifications. Additionally, in the Cairo dataset, meal-related activities like 'Breakfast', 'Lunch', and 'Dinner' are more accurately identified.

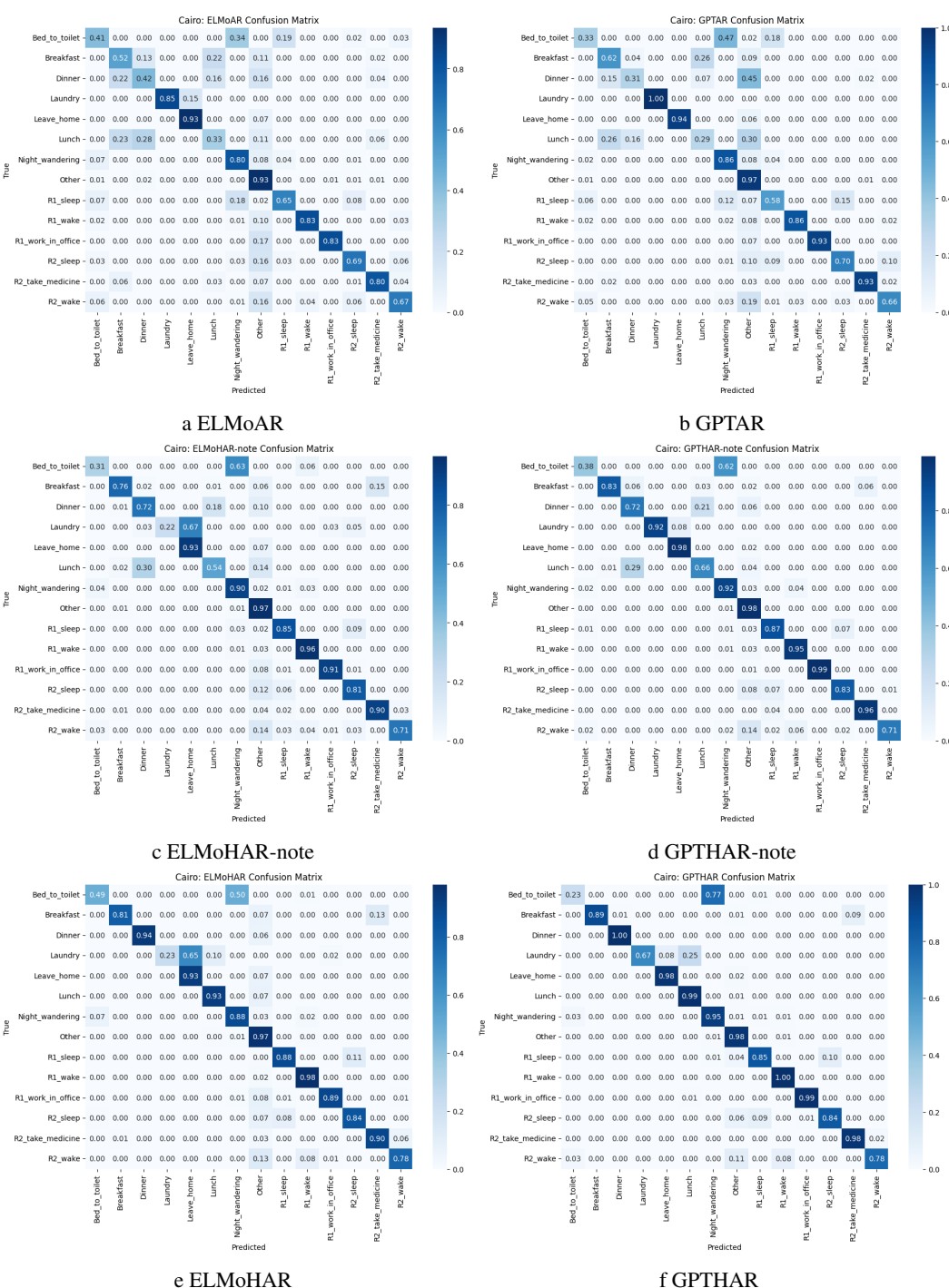

Figure 5: Confusion matrices per algorithm on the Cairo dataset.

-These findings highlight the effectiveness of our improved algorithms in reducing misclassifications.

# E    DATASETS DETAILS

This section provides detailed information on each dataset. It includes specifics such as the number of residents, the number of sensors utilized, the variety of activities recorded, and the duration of these recordings.

Table 11: Details of the three CASAS datasets

| Dataset | Aruba | Milan | Cairo |
|---|---|---|---|
| Residents | 1 | 1 + pet | 2 + pet |
| Number of Sensors | 39 | 33 | 27 |
| Number of Activities | 12 | 16 | 13 |
| Number of Days | 219 | 82 | 56 |

# F  EXPERIMENTS REPRODUCTION

This section presents the scores and results of our replication of the Liciotti et al. Liciotti et al. (2019) experiments. Table 12 below includes results from the original paper for the Milan and Cairo datasets, alongside our findings. It's important to note that the Aruba dataset had already been explored in the original study. We conducted our experiments 10 times, and the table reflects the average results of these repetitions. In line with the original paper, we employed a 3-fold cross-validation evaluation method for each dataset and regrouping the original dataset labels under meta-activities using the same activity remapping as in the original study. Additionally, we adhered to the same hyperparameters defined in the original paper to ensure consistency in our replication process. Our findings demonstrate that we achieved results closely similar to those in the original paper, confirming the accuracy of our implementation of the algorithm.

Table 12: Reproduction results of the Bi-LSTM architecture as proposed by Liciotti et al. Our replication of these experiments was conducted 10 times to ensure reliability and consistency of the results.

|  | Aruba | | Milan | | Cairo | |
|---|---|---|---|---|---|---|
|  | Liciotti et al. (paper) | Liciotti et al. (reproduce) | Liciotti et al. (paper) | Liciotti et al. (reproduce) | Liciotti et al. (paper) | Liciotti et al. (reproduce) |
| Accuracy | NA | 96.17% | 94.12% | 90.70% | 86.90% | 86.67% |
| Precision | NA | 92.73% | NA | 82.33% | NA | 79.67% |
| Recall | NA | 90.17% | NA | 77.00% | NA | 75.00% |
| F1 Score | NA | 91.17% | NA | 79.33% | NA | 76.33% |
| Balanced Accuracy | NA | 90.37% | NA | 76.88% | NA | 74.98% |
| Weighted Precision | NA | 96.30% | 94.00% | 90.33% | 86.67% | 86.33% |
| Weighted Recall | NA | 96.10% | 94.00% | 90.67% | 87.00% | 86.67% |
| Weighted F1 Score | NA | 96.03% | 94.00% | 90.33% | 86.67% | 86.33% |

