# OpenReview forum: "Generative Pretrained Embedding and Hierarchical Representation to Unlock Human Rhythm in Activities of Daily Living"
_ICLR.cc/2024/Conference — ICLR 2024 Conference Withdrawn Submission_

### Official Review · Reviewer_e66t · 2023-10-27

**Soundness:** 2 fair
**Presentation:** 3 good
**Contribution:** 2 fair
**Rating:** 3
**Confidence:** 5

**Summary:**

This paper presents an approach to human activity recognition in smart homes, that is using sensors integrated into a domestic environment to capture human activities. Said activities are then analyzed through a sequin sequential model that is based on sensor embeddings that utilize modeling approaches that are known from the domain of language analysis. The claimed innovation lies in the replacement of ELMO embeddings as they had have been used in previous work with GPT embeddings, and the introduction of contextual, hierarchical activity analysis. The experimental evaluation is based on standard benchmarks, i.e., the CASAS datasets and results are presented in form of balanced accuracies and comparisons are drawn to previous methods that were ELMO based, and a deeper dive into the effectiveness of hierarchical modeling is presented.

**Strengths:**

This paper operates in an interesting and very relevant application area: Human activity recognition in smart homes has many practical applications, for example, in home automation or in ambient assistant living scenarios. Activity recognition in such environments is inherently challenging due to the unconstrained environment, the noise, ambiguities in both sensor readings and annotations, and many other factors. As such, much progress still needs to be made and I applaud the authors for tackling such an important problem. The paper sets off from a relevant baseline and works with relevant benchmark datasets — as such the presented work in itself is relevant and has the potential to push beyond the state of the art.

**Weaknesses:**

Despite the general importance of the problem domain that this paper tackles, there are a number of weaknesses with this paper. First, the technical innovation is rather limited. The authors essentially replace one established sensor embedding (ELMO) with another one (GPT). Even the latter one has already been used in previous work (as cited by the authors — Takeda et al. 2023). The authors claim some additional technical improvement, namely the introduction of temporal context and hierarchical processing. While the former seems problematic because, in my opinion, in substantially limits the generalizability of the resulting models (I believe they are vey likely to overfit, which, alas has not been evaluated in detail), the latter seems interesting. The authors are right in stating that flat activity recognition has issues — especially when it comes to the analysis of concurrent activities. Yet, I am not convinced that the presented hierarchical approach would actually alleviate this problem in general as, for example, the incorporation of timestamps into the encoding / representation again limits generalizability substantially.
I am also concerned about the experimental evaluation — which needs to be described in more detail. From the description of the dataset splits I get the impression that at least some leakage is introduced during model training / hyper parameter tuning? Also: It is not clear to me what the basis for the evaluation is. The authors mention week-wise splits but are the actual continuous sensor readings processed or the pre-segmented activities? I suspect it is the latter (judging by the results on the CASAS datasets [I have substantial experience in working with these] — which is a problem because this would be a rather unrealistic evaluation.
There are also some issues with the presentation: For example it remains unclear what the authors mean by “rhythm of ADL” (which they aim to unveil).
Finally, I think the claim of causality in general is a bit of a stretch here. Yes, filling up an empty room requires the door to be opened and shut, but activities covered in CASAS do not generally follow this causality principle.

**Questions:**

1. How exactly are the datasets split for model training and hyperparameter tuning, as well as evaluation? Is there leakage?
2. Are you using pre-segmented activities or are you operating in continuous sensor data streams. Please provide evidence.
3. The improvements in recognition accuracy are barely significant — as per the table, and you only compare to one set of baseline methods. There are other models out there, why not comparing to them?
4. Why using such a rather exotic evaluation measure (balance accuracy) and not the regular macro F1 scores that one should use for such imbalanced datasets?

---

> ### Author Response · Authors · 2023-11-21
> **Answering the questions Q1 and Q2**
>
> Thank you for your insightful review of our paper and valuable suggestions, which have significantly contributed to the improvement of our work.
>
> > Q1-   split for model training and hyperparameter tuning, as well as evaluation? Is there leakage?
>
> In our study, we have split the datasets into separate sets for model training, hyperparameter tuning, and evaluation, ensuring the integrity of the process and avoiding  data leakage.
>
> Dataset Splitting Process:
>
> Initially, we divided the datasets into weekly segments. This approach helps maintain continuity between days, as some activities, like sleeping or nighttime wandering, can span across days.
> %These weekly segments are then further divided into training and test sets.
> A random 30 \% of these weeks are kept as the test set.
> From the remaining 70 \%, we subdivide each week into pre-segmented activities and use 80 \% for training and 20 \% for cross validation.
>
> Embedding Pre-training:
>
> For the pre-training of embeddings, we exclusively used the training and cross-validation sets.
> In the case of ELMo-based embedding, the training set's weekly segments are further split into pre-segmented activities. We allocated 20\% of these pre-segmented activities for early stopping and validation.
> For the GPT-based embedding, we applied a sliding window technique to create chunks from the weekly segments, before pre-segmenting into activities. Again, 20\% of these chunks were reserved for early stopping and validation.
>
> Cross Validation :
>
> For cross-validation, we evaluated the performance on the cross-validation set of the model trained on the training set.  There is a potential for leakage between the train and the validation phases, due to the overlap between the subsets used for the pre-trained embedding tasks and the validation set. This overlap might result in the embedding encountering some sensor sequences in both the training and validation phases.
>
> Test Results :
>
> Finally, the classifiers trained through this process are evaluated on the test set, which has not been used in any previous stages of pre-training, cross-validation, or classification tasks.
> This methodology has been carefully designed to preserve the integrity of the evaluation process and avoid the risk of data leakage for the test results.
>
> > Q2-  pre-segmented activities or  continuous sensor data streams.
>
> We are utilizing pre-segmented activities for the classification tasks and the training of ELMo embeddings. The nature of our approach with ELMo embeddings requires pre-segmentation. This method ensures that each segment corresponds to a distinct activity, allowing for more accurate training and classification.
>
> In contrast, our GPT-based embedding does not require pre-segmentation into activities. It utilizes chunks of continuous sensor data events, which do not necessarily correspond to complete activities. These chunks may contain a single activity, multiple activities, or even just a portion of an activity. Each chunk is a sequence of sensor events selected randomly, but the chronological order of these sensor events is maintained. This approach allows the GPT model to learn from a broader context, including partial and overlapping activities, providing a more nuanced understanding of the sensor data streams.

---

> ### Author Response · Authors · 2023-11-21
> **Answering questions Q3 and Q4**
>
> > Q3 - The improvements  are barely significant... There are other models out there, why not comparing to them?
>
> In response to your comment regarding the significance of our recognition accuracy improvements and the scope of our comparisons, we acknowledge the importance of a comprehensive comparative analysis. Initially, our comparison was focused on the ELMo-based approach, as it represented a recent benchmark in the field and had already been compared with a range of other methods in previous studies. This comparison provided a relevant and contemporary point of reference for our work.
>
> However, in light of your suggestion, we have broadened the scope of our comparative analysis. We have now included additional comparisons with other prominent models in the field in table 3 of section 4.2.1. Specifically, we have compared our approach with the work of Liciotti et al., which utilizes a SOTA LSTM-based methodology, and a CNN approach employing a Fully Convolutional Network (FCN) architecture. As Liciotti et al.'s algorithm has already demonstrated superiority over traditional machine learning approaches, including Naive Bayes, Conditional Random Field, LSTM and HMM, by proxy, our GPT-based algorithms thus outperform also Naive Bayes, Conditional Random Field, LSTM and HMM.
> This expanded comparison allows us to position our work within a wider context of existing methods and provides a more comprehensive evaluation of our model's performance.
>
> > Q4- Why  not the regular macro F1 scores that one should use for such imbalanced datasets?
>
> We initially chose balanced accuracy as our evaluation metric based on observation of classical accuracy use in multiple literature sources which might not fully represent performance in the context of imbalanced datasets. However, we acknowledge your point regarding the suitability of macro F1 scores for such datasets.
>
> In response to your feedback, we have revised our evaluation approach. We have replaced balanced accuracy with macro F1 scores in our analysis, recognizing that this metric provides a more comprehensive and relevant measure of performance for imbalanced datasets. Additionally, to enhance the transparency and detail of our evaluation, we have included confusion matrices for each dataset of main algorithms in the appendix. These matrices highlight the performance of each algorithm across different classes. Furthermore, we have provided tables in the appendix detailing the F1 scores for each dataset and activity, offering a granular view of each algorithm's performance.
>
> These amendments aim to present a more accurate and thorough evaluation of our model, aligning with standard practices in the field.

---

> ### Author Response · Authors · 2023-11-22
> **Answering to weaknesses (part 1)**
>
> > W1: the technical innovation is rather limited.
>
> Our approach introduces several innovative elements. While Takeda et al. (2023) utilized a GPT-based model primarily for predicting the next sensor event, our method extends its application to classification tasks. Additionally, unlike their work, we explore the integration of pre-trained embeddings from  actual continuous sensor readings, without pre-segmentation, whereas Takeda et al's GPT-2 module needs the label output by a separate classification module.
>
> >  W2: the introduction of temporal context ... substantially limits the generalizability of the resulting models
>
> The temporal aspect is crucial for activities of daily living, for instance in the medical perspective. It allows the detection of variations in behavior, such as schedule shifts or anomalies, which could indicate underlying health issues for the elderly.
> Moreover, we believe that incorporating time does not hinder generalization; rather, it aids in distinguishing activities with similar patterns that occur at different times of the day. This is particularly the case of activities like breakfast, lunch or dinner. The confusion matrices that we added in the annex B, show for the dataset Cairo that the three activities are confused for the algorithm without the timestamp embedding, when their confusion value drops to 0.00 or 0.01 when adding the timestamp encoding.
>
> For instance, for the version without timestamp encoding (GPTHAR-note where "note" stands for no time encoding), the confusion matrix for these 3 activities  on the Cairo dataset is :
>
> |  Activity    | Breakfast | Dinner     | Lunch |
> |--------------|-----------|------------|-------|
> | Breakfast    |     0.83  | 0.06       |  0.03 |
> | Dinner       |     0.0   |   0.72     |  0.21 |
> | Lunch        |     0.01  |    0.29    |   0.66|
>
>
> In contrast, for the version with timestamp encoding (GPTHAR), the confusion matrix for these 3 activities is :
>
> |  Activity    | Breakfast | Dinner     | Lunch |
> |--------------|-----------|------------|-------|
> | Breakfast    |  0.89     |     0.01   |  0.00 |
> | Dinner       |  0.00     |      1.00  |  0.00 |
> | Lunch        |  0.00     |      0.00  |  0.99 |
>
> The confusion matrix shows that the timestamp information can help discriminate activities.
> The issue of encoding time in time series is more general than for activity recognition, and has been investigated for instance in [1.2]. Our efforts proceed from the same question how to integrate the timestamp information in the time-series modelling. After testing positional encoding for the order of the signal, with absolute timestamp and with relative timestamp, we report in this article that the most efficient from our experimental results is by using the hour from the absolute timestamp.
>
>
> > W3: are the actual continuous sensor readings processed or the pre-segmented activities?
>
> As detailed in our response to Q1, for the embedding layer with GPT transformer decoder, we use the actual continuous sensor readings. In contrast, the ELMo embedding uses pre-segmented activities. This shows the improvement of our proposition, which not only replaces one established sensor embedding (ELMO) with another one (GPT), but enables the embedding to be pre-trained in a totally unsupervised manner, whereas ELMo needs pre-segmented data.
> Beyond the embedding layer with GPT or ELMo, however, for classification, we use pre-segmented activities. We will clarify this aspect in section 3.3.
>
>
> [1] Foumani, N.M., Tan, C.W., Webb, G.I. et al. Improving position encoding of transformers for multivariate time series classification. Data Min Knowl Disc (2023). https://doi.org/10.1007/s10618-023-00948-2
> [2] Yang, C., Chen, Y., Li, Z., Wang, X. (2023). Exploring the Effectiveness of Positional Embedding on Transformer-Based Architectures for Multivariate Time Series Classification. In: Yang, X., et al. Advanced Data Mining and Applications. ADMA 2023. Lecture Notes in Computer Science(), vol 14176. Springer, Cham. https://doi.org/10.1007/978-3-031-46661-8_3

---

> > ### Author Response · Authors · 2023-11-22
> > **Answering to weaknesses (part 2)**
> >
> > > W5:  the claim of causality in general is a bit of a stretch here
> >
> > We agree that our approach does not fully resolves the challenge of understanding causality, we posit that using a causality-focused design like the GPT architecture allows the algorithm to better highlight causal relationships between events.
> > Indeed for activity recognition, several approaches based on logic have been deployed to model the interrelationship between sensor activations and between activities. A review of logical modelling and reasoning algorithms to exploit logical knowledge representation for activity and sensor data modelling, and to use logical reasoning to perform activity recognition was described in [3]. Thus, not only a probablistic model but also a more logic-based relationship seem to underlie the structure of activities of daily living.
> > This perspective can enhance the understanding of activities and assist in filtering out noise from interleaved activities. By leveraging the inherent causal learning attributes of the GPT model, we aim to provide deeper insights into the sequential and causal nature of actions into the activities being analyzed. The dataset Cairo is an illustrated example were understanding some causality could help in interleaved activities recognition task.

---

### Official Review · Reviewer_wCG6 · 2023-10-31

**Soundness:** 3 good
**Presentation:** 3 good
**Contribution:** 2 fair
**Rating:** 6
**Confidence:** 4

**Summary:**

This paper present an approach to human activity recognition (HAR) from ambient sensors in smart home setting. Transformer decoder based pre-trained embedding is proposed, considering hierarchical sequential architecture and time encoding to refine the model. Three long-term activity recognition datasets are benchmarked with promising results.

**Strengths:**

Paper provides novel combination of existing ideas (pre-trained transformer (GPT-like design, bi-directional LSTM) to build hierarchical model. Based on empirical evaluation it shows the usefulness of the hierarchical modelling of activities. Building blocks are quite-well justified and results are promising; improving some of the issues in previous approach.

**Weaknesses:**

Paper is application oriented in quite well-defined domain, and is an incremental improvement to a previous study. It lacks "basic" baseline other than GPT/LLM-style of model in comparison. Also, there are some stability issues which might be tackled with the normalisation layer, but that has not been evaluated in practice.

**Questions:**

- How would "basic" baseline, i.e. hierarchical HMM compared to deep learning models (in this setting)?
- It would be useful to evaluate further the stability issue of hierarchical models (e.g., using normalisation layer)
- It would be useful to show the confusion matrix of different activities and which are most difficult to discriminate
- Can you discuss about sensor data processing with symbolic representation and how that effect HAR? E.g., continuous temperature
measurements are now transformed to symbolic labels, compared to more traditional sensor signal processing approaches which uses
directly the numerical sensor values.

---

> ### Author Response · Authors · 2023-11-22
> **Answering the questions Q1**
>
> Based on your suggestion, we have now expanded our study to include comparisons with other methods in table 3 of section 4.2.1. Specifically, we conducted a comparison with another method than sequence models : a CNN based approach, more precisely a FCN [2] model.
> Additionally, we compared our approach against the work of Liciotti et al.[1], a SOTA LSTM-based method. It's important to note that Liciotti et al.'s algorithm has already demonstrated superiority over traditional machine learning approaches, including Naive Bayes, Conditional Random Field, LSTM and HMM, in [1].
>
> Our findings indicate that both the FCN and Liciotti et al.’s LSTM-based approach are outperformed by far by all versions of ELMo-based and GPT-based methods. The F1-scores more than double for ELMoAR, GPTAR, ELMoHAR-note and GPTHAR-note.
> By proxy, our GPT-based algorithms thus outperform also Naive Bayes, Conditional Random Field, LSTM and HMM.
>
>
> |                         |Aruba | Milan| Cairo|
> |--------------|-----------|------------|-------|
> |FCN [2]                                | 33.10 ± 2.23 |  15.10  ± 1.52  | 7.60± 2.46.    |
> |Liciotti et al.  [1]                   | 32.00 ±1.56  | 17.40 ± 2.07    | 26.60 ±3.24.  |
> |ELMoAR (Window 60)         | 84.80 ± 1.99 | 70.80 ±0.92     |  70.50 ±1.43  |
> |GPTAR (8 Heads 3 Layers)  | 86.10 ± 1.20 | 70.80 ± 1.40    | 73.20 ±1.99   |
> |ELMoHAR-note                   | 88.10 ± 1.20 | 77.40  ± 1.65   | 75.90  ± 2.88 |
> |GPTHAR-note                     | 87.30 ± 2.98  | 79.90 ± 1.52   | 84.80 ± 1.81  |
>
>
> To verify the accuracy of our implementation of the comparative algorithm from article [1], we replicated their methodology, details can be found in Annex F. This process involved relabeling activities of the same type and employing 3-fold cross-validation across each dataset. Our findings were consistent with the results reported in their paper.
>
> This comprehensive comparison framework not only adheres to your suggestions but also reinforces the effectiveness of our proposed GPT-based model.
>
> [1] Liciotti, D., Bernardini, M., Romeo, L., and Frontoni, E. (2019). A Sequential Deep Learning Application for Recognising Human Activities in Smart Homes. Neurocomputing.
>
> [2] Bouchabou, D., Nguyen, S. M., Lohr, C., Leduc, B., and Kanellos, I. (2021). Fully convolutional network bootstrapped by word encoding and embedding for activity recognition in smart homes. Deep Learning for Human Activity Recognition: Second International Workshop, DL-HAR 2020, Held in Conjunction with IJCAI-PRICAI 2020, Kyoto, Japan, January 8, 2021, Proceedings 2(111--125).

---

> ### Author Response · Authors · 2023-11-22
> **Answering the questions Q2**
>
> Addressing the stability issues observed in hierarchical models is indeed crucial. Evaluating the impact of a normalization layer, as you suggested, is highly relevant. Although this aspect was not covered in our current study, we recognize its potential to enhance model performance. We plan to explore this in future iterations of our research, focusing on how various normalization techniques could mitigate stability issues in hierarchical deep learning models. Furthermore, we believe that increasing the number of neurons in the hidden layers could resolve stability issues by enhancing the model's representational capabilities. This hypothesis is inspired by our observations of performance variations across different classes during training. We noticed differences in the model's focus on specific activities, which seemed to shift from one training session to another. By adding more neurons, we anticipate that the model will have sufficient representational capacity to maintain a consistent focus across all activities.

---

> ### Author Response · Authors · 2023-11-22
> **Answering the questions Q3**
>
> Thank you for your insightful recommendation regarding the inclusion of confusion matrices.
>
> We have added the average confusion matrices for each algorithm and dataset in Annex C. This addition aims to facilitate a clearer understanding of the performance improvements achieved by each algorithmic increment.
>
> For example, in the Aruba dataset, activities such as 'Washing Dishes', 'Meal Preparation', 'Enter Home', and 'Leave Home' show improved classification accuracy. In the Milan dataset, 'Eve Med' and 'Morning Meds' demonstrate a notable reduction in misclassifications. Similarly, in the Cairo dataset, meal-related activities like 'Breakfast', 'Lunch', and 'Dinner' are identified with higher accuracy.
>
> To illustrate, consider the Cairo dataset in the version without timestamp encoding (GPTHAR-note, where "note" signifies no time encoding). The confusion matrix for 'Breakfast', 'Dinner', and 'Lunch' is as follows:
>
> |  Activity    | Breakfast | Dinner     | Lunch |
> |--------------|-----------|------------|-------|
> | Breakfast    |     0.83  | 0.06       |  0.03 |
> | Dinner       |     0.0   |   0.72     |  0.21 |
> | Lunch        |     0.01  |    0.29    |   0.66|
>
> Contrastingly, for the version with timestamp encoding (GPTHAR), the confusion matrix for these activities is:
>
> |  Activity    | Breakfast | Dinner     | Lunch |
> |--------------|-----------|------------|-------|
> | Breakfast    |  0.89     |     0.01   |  0.00 |
> | Dinner       |  0.00     |      1.00  |  0.00 |
> | Lunch        |  0.00     |      0.00  |  0.99 |

---

> ### Author Response · Authors · 2023-11-22
> **Answering the questions Q3**
>
> Thank you for raising this important question. Transforming continuous sensor data into symbolic labels marks a substantial departure from conventional sensor signal processing techniques, particularly in the realm of Human Activity Recognition (HAR). This strategy simplifies complex continuous data into symbolic forms that are more manageable and interpretable. For example, in our research, we convert temperature sensor readings into symbols. This method proves especially useful in smart home environments, where temperature sensors typically operate within a specific range and send data when threshold changes occur. Consequently, these readings result in event-based values rather than continuous streams, making symbolic representation a more appropriate and effective approach. However, it's worth noting that we have not yet had the opportunity to apply this method to real continuous data from sensors like power meters as our current studied datasets doesn't contains such sensors values.
>
> When considering real continuous data from sensors, we anticipate adapting our approach. Instead of transforming all continuous values into symbols, we plan to use symbols to represent specific threshold crossings or distinct events. For instance, the activation or deactivation of an appliance generates a unique signal pattern in power signals. We aim to symbolically represent these patterns with symbols that convey changes in the appliance's status, based on the detection of appliance patterns in the raw signal.

---

> > ### Comment · Reviewer_wCG6 · 2023-11-23
> > **Response to rebuttal**
> >
> > Thank you for the response to clarify some of my concerns. Extended evaluation indeed shows some promising practical results of hierarchical modelling (e.g., against the baseline, confusion matrices etc.), which can be seen as a main contribution. The  technical contribution and novelty is a bit limited, combining already existing approaches. Furthermore, stability of hierarchical model could have been studied more carefully in practice. I'll keep my original score.

---

### Official Review · Reviewer_dgfF · 2023-11-01

**Soundness:** 2 fair
**Presentation:** 2 fair
**Contribution:** 2 fair
**Rating:** 5
**Confidence:** 3

**Summary:**

The paper proposes a multi-time scale architecture aiming to leverage a wider temporal context in a multi-time scale manner. The core problem to solve is classifying sensor event sequences. The temporal order of the sequences are important for reasoning in this application.

**Strengths:**

The primary contribution of this paper is to leverage Transformer decoder for sensor embedding and hierarchical architecture design.
These techniques appear to be adaptation of existing methodologies for this domain which hasn't been explored before. The core *technical* contribution could have been a bit more.

**Weaknesses:**

I have a question and concern about the presentation of the paper. All the tables look like ablation results and collection of different baselines. The entries in the tables aren't clear which one is hierarchical vs which one is not. The captions need to be improved and self-explanatory. I am still confused what's the proposed method? Is the "GPTHAR+Time-encoding" in Table 5? OR, this paper is a review paper. It needed a second read to understand the differences.

**Questions:**

They have reported only the balanced accuracy metric. It's good to check other metrics such as table 6 or 7 of previous SOTA paper: https://arxiv.org/pdf/2111.12158.pdf
Are these datasets long-tailed? is the balanced accuracy increasing at the cost of accuracy?
The annexture provided some of those additional metrics. I'd suggest highlighting the best class per method would be good.

---

> ### Author Response · Authors · 2023-11-22
> **Answering to Questions**
>
> Thank you for your valuable feedback.
>
> Regarding your comment on our use of metrics, we acknowledge your suggestion about exploring additional metrics as exemplified in tables 6 and 7 of the SOTA paper (https://arxiv.org/pdf/2111.12158.pdf). In response, to your comments on our choice of metrics, we have updated our analysis to include macro F1 scores instead of balanced accuracies in our main table to provide a more comprehensive understanding of our results. We agree that a range of metrics can offer a richer insight. Due to paper length constraints, the main body of the paper will feature only the F1 scores.  However, detailed tables in Annex A will provide a complete breakdown of all metrics for each algorithm across the three datasets, as presented in tables 6 and 7 of the SOTA paper (https://arxiv.org/pdf/2111.12158.pdf).
>
> Regarding your concern about the datasets being long-tailed and the trade-off between balanced accuracy and accuracy, we have conducted a more thorough examination. In the annex, we present additional metrics to illuminate this issue. We have also highlighted the best-performing class for each method in our results. To further enhance clarity, confusion matrices for each class are included in the appendix. These additions aim to more effectively address the nuances of dataset characteristics and their implications on our evaluation metrics.
>
> We believe these revisions will substantially improve the clarity and thoroughness of our analysis and are grateful for your constructive suggestions.

---

> ### Author Response · Authors · 2023-11-23
> **Answering to weaknesses**
>
> Thank you for your comment. We have improved the presentation of the paper to make clearer what is the proposed method : the algorithm using GPT transoformer decoder with time-encoding and a hierarchical architecture. We re-named our proposed method GPTHAR.
>
> Now,  the algorithms have been renamed as :
> - GPTHAR : using GPT transformer decoder with time-encoding and a hierarchical architecture.
> - GPTAR-note (for no temporal encoding) : uses GPT transoformer decoder and a hierarchical architecture, but  without timestamp information .
> - GPTAR : using GPT transformer decoder for embedding in a single-level architecture, without timestamp information.
>
> - ELMoHAR : using ELMo with time-encoding and a hierarchical architecture.
> - ELMoAHR-note (for no temporal encoding) : uses ELMo and a hierarchical architecture, but  without timestamp information .
> - ELMoAR : using  ELMo for embedding in a single-level architecture, without timestamp information.

---

### Official Review · Reviewer_v1qx · 2023-11-02

**Soundness:** 3 good
**Presentation:** 3 good
**Contribution:** 2 fair
**Rating:** 5
**Confidence:** 3

**Summary:**

The paper presents an approach for temporal human activity detection in smart homes using GPT-based hierarchical model. The authors test their method on 3 datasets

**Strengths:**

The authors focus an important problem in the context of smart buildings. Activity detection using efficient machine learning methods help achieve occupant comfort and energy efficiency if these inputs are fed to building control mechanism.

**Weaknesses:**

1. The authors have not covered more on the types of activities captured in the datasets, and their importance in smart homes, particularly from the perspective of occupant comfort and energy efficiency.
2. The number of sensors used to collect data seems a lot. In practice, its not practical to have so many sensors in a home collecting information. The authors should try some benchmarking on a subset of sensors if the dataset permits.
3. How will a sensor fusion approach work in this scenario?
4. What are the motivations behind hierarchical approach?
5. For Milan and Cairo, the temporal method might not be effective since the number of days in the experiment is less.

**Questions:**

See the question above.

---

> ### Author Response · Authors · 2023-11-21
> **Response to the questions**
>
> We thank you for outlining the importance of this field of research.
>
> >  Q1: the types of activities...
>
> Indeed, several typologies of activities have been proposed to capture their importance from the perspectives of comfort, energy efficiency or health.
> For instance they have helped set up the Index of Independence in Activities of Daily Living (ADL), now in frequent use in rehabilitation settings, for applications such as the prevention of disability and maintenance of rehabilitation gains in the ageing person [1]. For this medical perspective, Katz et al. [1] defined a list of basic and necessary activities to analyse.
>
> However, in this study, we keep the original labels of activities for each dataset, to showcase the performance of our algorithm in the general case with arbitrary labels, and to avoid biasing our results in a specific case of typology of activities.
>
> > Q2: The number of sensors... benchmarking on a subset of sensors
>
> Thank you for raising one of the main issues why the current models learned from datasets can not yet be deployed in users' homes. It is indeed also our stance that most datasets might not be realistic as they use a higher number of sensors than what are currently sold for households. This impacts not only the redundancy of data due to the number of sensor activations in the data stream, but also the nature and sensor signals as it changes how and where the sensors can be positioned. This is why, in partnership with a company specialised in IoT sensors for smart homes, we designed a set of sensors that is more realistic to the observed sales and have been conducting data collections in 3 homes with one occupant for 4 months. We will report  in future work our findings on that dataset, and will also  explore with public datasets, how benchmarking on a subset of sensors affects our activity recognition algorithms.
>
> > Q3 : How will a sensor fusion approach work ?
>
> While sensor fusion approaches have been proposed for activity recognition based on a network of IoT in smart homes, their approach is fundamentally different. In general, instead of using the raw data stream as a event-triggered multi modal time series, these approaches re-sample them as a multi-dimensional regular time-series. Instead, our approach takes the event-triggered multi modal time series without temporal resampling, and embeds all sensors data in the same embedding space.
>
> > Q4 : the motivations behind hierarchical approach?
>
> The theoretical motivations behind our study of a hierarchical approach is to understand how temporal sequential relations in complex actions can be represented and modelled, and activities of daily living can be seen as the most complex actions human do on a regular basis, due to their variability and compositionality.
> As outlined in section 5.2 of the related works, we are motivated by studies in neuroscience [2], as well as the behavioural psychology [3], which proposed a hierarchical organisation of actions.
>
> While this idea has been implemented in hierarchical reinforcement learning for simple tasks, we wish here to investigate for tasks as complex as activities of daily living, if a hierarchical model is necessary, as proposed in hierarchical models such as in [4] or hierarchical hidden Markov models [5].
>
> We added a paragraph in the introduction to clarify our motivation.
>
> > Q5 : For Milan and Cairo, the temporal method might not be effective...
>
> We are not sure we could understand the meaning of your questions. Could you please clarify your questions ?
>
> To our knowledge, the number of days in each dataset does not affect the algorithm per se, as the data used as input is divided by a week each for pre-training the GPT embedding, and divided further by activity sequence for the classification. The only impact of the number of days available is the number of data our neural network can train on. Indeed, we have for Milan and Cairo only few data, even compared to the Aruba dataset. This is why we used 3 fold cross-validation for our hyper-parameter search.
>
> [1] S. Katz, T. D. Downs, H. R. Cash, and R. C. Grotz, “Progress in devel- opment of the index of adl,” The gerontologist, vol. 10, no. 1 Part 1, pp. 20–30, 1970
>
> [2] Scott T. Grafton and Antonia F. de C. Hamilton. Evidence for a distributed hierarchy of action representation in the brain. Human Movement Science, 26(4):590–616, 2007.
>
> [3] Maria K Eckstein and Anne G E Collins. How the mind creates structure: Hierarchical learning of action sequences. In Cognitive Science Society (ed.), CogSci Conference of the Cognitive Science Society, volume 43, pp. 618–624, 2021
>
> [4] Xin Hong, Chris Nugent, Maurice Mulvenna, Sally McClean, Bryan Scotney, and Steven Devlin. Evidential fusion of sensor data for activity recognition in smart homes. Pervasive and Mobile Computing, 5(3):236 – 252, 2009.
>
> [5]Parviz Asghari, Elnaz Soelimani, and Ehsan Nazerfard. Online human activity recognition employing hierarchical hidden markov models, 2019.

---

### Author Response · Authors · 2023-11-23
**Summary of the main changes**

Dear reviewers, we earnestly  thank you for the valuable feedback and discussions. We are convinced it helped us improve the quality of our paper.
We submitted a new paper’s version to address the most pressing feedback :

- **Presentation of the proposed method** : We have improved the presentation of the paper to make clearer what is the proposed method : the algorithm using GPT transformer decoder with time-encoding and a hierarchical architecture. We re-named the algorithms proposed and in the ablation studies:

    - GPTHAR : using GPT transformer decoder with time-encoding and a hierarchical architecture.

    - GPTAR-note (for no temporal encoding) : uses GPT transoformer decoder and a hierarchical architecture, but without timestamp information .

    - GPTAR : using GPT transformer decoder for embedding in a single-level architecture, without timestamp information.

    - ELMoHAR : using ELMo with time-encoding and a hierarchical architecture.

    - ELMoAHR-note (for no temporal encoding) : uses ELMo and a hierarchical architecture, but without timestamp information .

    - ELMoAR : using ELMo for embedding in a single-level architecture, without timestamp information.

- **Comparing with different baselines**: we conducted a comparison with another method than sequence models : a CNN based approach, more precisely a FCN [2] model. Additionally, we compared our approach against the work of Liciotti et al., a SOTA LSTM-based method. It's important to note that Liciotti et al.'s algorithm has already demonstrated superiority over traditional machine learning approaches, including Naive Bayes, Conditional Random Field, LSTM and HMM, in Liciotti et al (2019).
Our findings indicate that both the FCN and Liciotti et al.’s LSTM-based approach are outperformed by far by all versions of ELMo-based and GPT-based methods. The F1-scores more than double for ELMoAR, GPTAR, ELMoHAR-note and GPTHAR-note. By proxy, our GPT-based algorithms thus outperform also Naive Bayes, Conditional Random Field, LSTM and HMM.

-**Evaluation measures** : We have changed the main metric of the algorithms to F1-score, and have added other metrics (accuracy, balanced accuracy ... ) and the confusion matrices in the Annex B and D.